# Genetic Deletion of HLJ1 Does Not Affect Blood Coagulation in Mice

**DOI:** 10.3390/ijms23042064

**Published:** 2022-02-13

**Authors:** Man-Chen Hsu, Wei-Jia Luo, Bei-Chia Guo, Chia-Hui Chen, Po-An Hu, Yi-Hsuan Tsai, Kang-Yi Su, Tzong-Shyuan Lee

**Affiliations:** 1Department of Physiology, College of Medicine, National Taiwan University, Taipei 10051, Taiwan; f07441013@ntu.edu.tw (M.-C.H.); d07441002@ntu.edu.tw (B.-C.G.); chiahuichen1993@ntu.edu.tw (C.-H.C.); d07441003@ntu.edu.tw (P.-A.H.); angeltsai1202@gmail.com (Y.-H.T.); 2Department of Clinical Laboratory Sciences and Medical Biotechnology, College of Medicine, National Taiwan University, Taipei 10051, Taiwan; f04424006@ntu.edu.tw

**Keywords:** HLJ1, DNAJB4, platelet, coagulation

## Abstract

HLJ1 (also called DNAJB4) is a member of the DNAJ/Hsp40 family and plays an important role in regulating protein folding and activity. However, there is little information about the role of HLJ1 in the regulation of physiological function. In this study, we investigated the role of HLJ1 in blood coagulation using wild-type C57BL/6 mice and HLJ1-null (HLJ1^-/-^) mice. Western blot analysis and immunohistochemistry were used to assess the expression and distribution of HLJ1 protein, respectively. The tail bleeding assay was applied to assess the bleeding time and blood loss. A coagulation test was used for measuring the activity of extrinsic, intrinsic and common coagulation pathways. Thromboelastography was used to measure the coagulation parameters in the progression of blood clot formation. The results showed that HLJ1 was detectable in plasma and bone marrow. The distribution of HLJ1 was co-localized with CD41, the marker of platelets and megakaryocytes. However, genetic deletion of HLJ1 did not alter blood loss and the activity of extrinsic and intrinsic coagulation pathways, as well as blood clot formation, compared to wild-type mice. Collectively, these findings suggest that, although HLJ1 appears in megakaryocytes and platelets, it may not play a role in the function of blood coagulation under normal physiological conditions.

## 1. Introduction

Blood coagulation is a very delicate process including a series of rigorously controlled cellular and molecular events that ultimately produces a stable blood clot [1]. Physiologically, coagulation begins almost immediately after the vascular endothelial injury. Exposing blood to the subendothelial space triggers two processes, namely changes in platelets and the exposure of subendothelial tissue factor to plasma factor VII, which ultimately leads to the formation of cross-linked fibrin [2,3]. Platelets immediately form a plug at the injury site, and then blood coagulation factors other than factor VII simultaneously respond in a cascade to form fibrin strands to strengthen the platelet plug and thereby prevent the loss of blood [2,3,4]. It has been documented that coagulation pathways can be divided into “intrinsic” and “extrinsic” routes initiated by factor XII (FXII) and FVIIa/tissue factor (TF), respectively. These two routes are integrated into the downstream “common” pathway at the FXa/FVa (prothrombinase) complex [5]. Pathologically, misfolded protein aggregates stimulate the activation of FXII but without inducing coagulation [6]. For example, the aggregated amyloid β peptide (Aβ) activates FXII in vitro, and the level of activated FXII is elevated in patients with Alzheimer’s disease (AD) and causes coagulation dysfunction [7]. Chaperones are a family of proteins that play a crucial role in the conformational folding or unfolding of proteins and assembly or disassembly with intracellular macromolecular structures [8,9]. Several lines of evidence demonstrate that chaperones are also important in regulating the activity of coagulation factors [10,11]. However, the molecular mechanism underlying the regulation of chaperones in blood coagulation is not fully understood.

HLJ1, an endogenous Src inhibitor, is also a member of the human heat shock protein (HSP) 40 subfamily and acts as a tumor suppressor in the pathogenesis of lung, colon, liver and gastric cancers [12,13,14]. HLJ1 inhibits cell invasion and impedes cell cycle progression by regulating the p21^waf1^ pathway in hepatocellular carcinoma and suppresses lung cancer progression by inhibiting Src activity [15,16]. Under physiological conditions, HSPs serve as molecular chaperones and regulate the stabilization of unfolded proteins [17]. In mammals, HSPs have been divided into six subfamilies according to their molecular size: HSP100, HSP90, HSP70, HSP60, HSP40 and small HSPs [18]. In addition to the relation to heat shock, HSPs are involved in the regulation of cellular functions upon stimulation with stimuli such as nutrient deprivation, oxidative stress and pathogen infection [19,20,21]. Recent studies have identified the role of HSPs in the blood coagulation of mammals [22,23]. For example, HSP72 promotes platelet aggregation, and HSP20 plays a vital role in coagulation function [24,25]. While the role of HLJ1 in cancer biology has been extensively investigated, its significance in blood coagulation is still elusive.

Given the physiological function of HSPs in protein folding and blood coagulation, we aimed to investigate the role of HLJ1 in blood coagulation in mice. We first investigated the expression of HLJ1 in bone marrow and plasma and then examined whether the genetic deletion of HLJ1 in mice affects blood coagulation. Our findings suggest that HLJ1 is expressed in megakaryocytes and platelets; however, genetic loss of HLJ1 function did not change the coagulation time, blood loss and the parameters of blood clot formation compared to wild-type (WT) mice.

## 2. Results

### 2.1. HLJ1 Is Expressed in the Bone Marrow and Plasma of Mice

To clarify the potential role of HLJ1 in regulating blood coagulation, we used WT and HLJ1^-/-^ mice as our animal models. We first examined the effect of the genetic deletion of HLJ1 on body weight and tissue weight in mice, and our results showed that loss of HLJ1 function increased the body weight, heart weight and white adipose tissue (WAT) weight; however, there was no difference in the weights of liver, spleen, lung, kidney, brown adipose tissue (BAT), gastrocnemius muscle and brain (Table 1). Moreover, there was no difference in the peripheral blood counts of WT and HLJ1^-/-^ mice (Table 2).

We next examined the expression of HLJ1 in the plasma and bone marrow of mice. The results showed that HLJ1 protein appeared in the platelet-rich plasma of WT mice as a soluble form; however, it was undetectable in that of HLJ1^-/-^ mice (Figure 1A). We then isolated platelets from plasma and found that no soluble form of HLJ1 was detected in platelet-poor plasma; in contrast, the protein expression of HLJ1 in plasma was mainly restricted in platelets, as evidenced by the expression of HLJ1 being co-localized with CD41, the marker for platelets and megakaryocytes (Figure 1B,C,E). In addition, we found that the protein expression of HLJ1 in bone marrow mainly appeared in megakaryocytes and platelets, as evidenced by the expression of HLJ1 being co-localized with CD41 in the bone marrow of WT mice (Figure 1D,F). 

### 2.2. The Role of HLJ1 in the Bleeding Time and Blood Loss in Mice

To evaluate the effect of HLJ1 on bleeding time and blood loss, we used *HLJ1^-/-^* mice as our animal model. The results showed that the genetic disruption of HLJ1 did not affect the bleeding time (Figure 2A,B) and blood loss (Figure 2C,D) compared to WT mice. 

### 2.3. The Role of HLJ1 in the Activity of Blood Coagulation in Mice

To investigate whether HLJ1 plays a role in regulating the blood coagulation response, we measured the coagulation parameters in WT mice and HLJ1^-/-^ mice. The results showed that the genetic deletion of HLJ1 in mice had no effect on the activity of intrinsic and extrinsic time (PT and aPTT, Figure 3A,B), the content of fibrinogen (FIB) and the time to fibrin clot formation (fibrinogen activity test) as compared to WT mice (Figure 3C,D). 

We further applied thromboelastography (TEG) to examine the role of HLJ1 in the dynamic real-time reaction of coagulation. The results demonstrated that there was no difference in the TEG parameters, including *R*, the time to formation of initial fibrin threads; *K*, the time until the clot reaches a certain strength; α angle, the rapidity with which the clot forms; maximum amplitude (MA), the clot’s maximum strength; amplitude (A), the end time amplitude; and the coagulation index, the overall clotting activity of HLJ1^-/-^ mice compared to WT mice (Figure 4A–H). 

## 3. Discussion

HLJ1, a chaperone protein, plays a crucial role in regulating protein folding stability and quality control [12,13,14]; however, the biological significance of HLJ1 in blood coagulation has not been fully investigated yet. In this study, we used WT and HLJ1^-/-^ mice as our in vivo model and characterized the potential effect of HLJ1 on blood coagulation. Our results suggest that HLJ1 indeed appeared in the megakaryocytes and platelets of the bone marrow and plasma of WT mice. Platelets are known to play an important role in regulating hemostasis, inflammation and wound healing and are thus implicated in the pathogenesis of human diseases under certain conditions [26,27]. However, the genetic disruption of HLJ1 did not change the platelet count of peripheral blood, the activity of blood coagulation and the formation of blood clots as compared to WT mice. Based on these observations, we concluded that HLJ1, a member of the HSP40 subfamily, is not involved in the activation of the coagulation system. Nevertheless, further investigations regarding the role of HLJ1 in the biological functions of platelets other than hemostasis are warranted. Whether HLJ1 plays an important role under specific stress induction should be further investigated.

Increasing evidence suggests that HSPs are expressed in platelets and participate in the regulation of platelet function and coagulation function [28,29]. For instance, Li et al. reported that HSP20 plays a vital role in coagulation function and decreased HSP20 might contribute to the pathogenesis of pre-eclampsia [25]. Zhu et al. reported that phosphorylation of HSP27 by thrombin is crucial for cytoskeletal rearrangements in the process of platelet activation [28]. HSP70 regulates platelet integrin activation, granule secretion and aggregation [29]. Suzuki et al. demonstrated that HSP72 promotes activator-induced platelet aggregation [24]. Very recently, Jackson et al. (2020) reported that the inhibition of HSP40 and HSP90 by pharmacological antagonists abolishes the dense granule release, thromboxane synthesis and aggregation in thrombin-treated platelets [30]. The evidence from these studies indicates that HSPs play an important role in platelet function and coagulation function; however, their regulatory mechanism is not fully understood. To this end, further investigation to delineate the molecular mechanisms of HSPs in platelet activation and blood coagulation is warranted.

To our knowledge, information about the role of HLJ1 in human diseases is limited. HLJ1 reportedly serves as a tumor suppressor and inhibits the proliferation, migration and invasion of cancer cells by regulating the STAT1/p21 pathway and could thus be a biomarker for cancers [15]. However, its role in other human diseases is largely unknown. In this study, we found that genetic deletion of HLJ1 caused a significant increase in body weight, which could be attributed to the increase in heart weight and WAT weight, suggesting that HLJ1 may play a key role in the regulation of heart diseases and metabolic diseases. Several lines of evidence indicate that the level of HSPs is correlated with the pathogenesis of cardiac hypertrophy [31,32,33,34]. Kumarapeli et al. have reported that the small HSP αB-crystallin (CryAB) is critical for normal cardiac function, and overexpression of CryAB suppresses pressure-overload-induced cardiac hypertrophy and delays the development of cardiac failure [31,32]. Nevertheless, whether HLJ1 is involved in the pathogenesis of cardiovascular diseases and its molecular regulation are unclear. To this end, further investigations for clarifying the role of HLJ1 in cardiovascular diseases and the underlying mechanisms are warranted.

Importantly, HSPs are involved in the pathogenesis of lipid disorders, including obesity, non-alcoholic fatty liver disease (NAFLD), insulin resistance and atherosclerosis [33,34,35,36,37,38,39,40,41,42]. Di Naso et al. reported that HSP70-mediated anti-inflammation is impaired in obese patients, which contributes to the development of NAFLD [35]. Chung et al. demonstrated that HSP72 protects against obesity-induced insulin resistance by blocking inflammation [36]. Circulating HSP70 acts as an inflammatory mediator and accelerates the progression of atherosclerosis [37]. Moreover, Gungor et al. reported that nuclear HSP70 has a cholesterol-lowering effect by activating the liver X receptor pathway [38]. The findings from these studies suggest that HSPs play an important role in the regulation of lipid metabolism and related metabolic diseases. This notion is further supported by our findings that genetic deletion of HLJ1 causes an increase in WAT weight. However, further investigations for delineating whether HLJ1 is involved in metabolic diseases and thermogenesis and its underlying molecular mechanism are required. 

Nevertheless, our study contains several limitations. HSPs are a large family of evolutionarily conserved molecular chaperones that regulate protein folding and maintain the protein structure and function in response to pathophysiological and environmental stimuli [18,43,44]. In this study, however, we did not use disease models to mimic the pathology of diseases and delineate the potential role and mechanism of HLJ1 in blood coagulation. Moreover, we do not have clinical data to support our observations from in vivo studies. Therefore, further pathological studies or clinical trials describing the implications of HLJ1 in coagulation disorders induced by various diseases are warranted. 

In conclusion, this study clarifies the relationship between HLJ1 and blood coagulation. Although HLJ1 is expressed in platelets and megakaryocytes, it has no role in the regulation of blood coagulation under physiological conditions; in contrast, HLJ1 may play a certain role in the regulation of cardiac function and lipid metabolism. Here, information should be provided for a better understanding of the function of HLJ1 in platelet biology and blood coagulation.

## 4. Materials and Methods

### 4.1. Reagents

Rabbit antibodies for β actin and HLJ1 (13064-1-AP) were obtained from Proteintech (Rosemont, IL, USA). Mouse antibody for CD41 (553847) was obtained from BD Bioscience (San Jose, CA, USA). NH_4_OH solution was obtained from PanReac AppliChem (Darmstadt, Germany). Prothrombin time (PT) activated partial thromboplastin time (aPTT) and fibrinogen (FIB) reagents were purchased from TECO Medical Instruments (Neufahrn, NB, Germany).

### 4.2. Animal

This study conformed to the Guide for the Care and Use of Laboratory Animals (Institute of Laboratory Animal Resources, Eighth Edition, 2011), and all animal experiments were approved by the Animal Care and Utilization Committee of the College of Medicine, National Taiwan University (20210257). HLJ1^-/-^ C57BL/6 mice were generated using the gene targeting strategy. Specifically, exon2 of mouse HLJ1 was deleted by traditional homologous recombination in embryonic stem cells. HLJ1^-/-^ mice were generated, and the polymerase chain reaction (PCR) of genomic DNA was performed to confirm the HLJ1^-/-^ genotype. PCR was performed with the following primers: HLJ1 (forward): 5′-GGC TTG CTG TCT AAG GTG ATG-3′, HLJ1 (reverse): 5′-ACG TTC AAA GCT ATC ATA GTC-3′. Primers were used at 95 °C for 5 min, followed by 40 cycles at 95 °C for 30 s, 62 °C for 30 s, 72 °C 30 s and additional extension at 72 °C for 5 min. Male wild-type (WT) C57BL/6 mice and HLJ1^-/-^ mice from the Laboratory Animal Center of the National Taiwan University College of Medicine (Taipei, Taiwan) were housed in barrier facilities on a 12-h light/12-h dark cycle. Five mice were group-housed per cage and fed with a regular chow diet containing 4.5% fat by weight and 0.02% cholesterol (Newco Distributors, Redwood, CA, USA). Two-month-old WT and HLJ1^-/-^ mice were euthanized with CO_2_ after measuring the body weight of mice. Heart, liver, spleen, lung, kidney, muscle, WAT and BAT were isolated and weighed. Plasma was isolated from citrated blood (3.2% sodium citrate, 1:10) by centrifugation for 10 min at 1500 g and 4 °C and then stored at −80 °C until further analysis.

### 4.3. Immunohistochemistry

Femur and tibia were isolated and fixed with formalin. After decalcification, the samples were embedded in paraffin. The blocks were then cut into 8 μm sections and subjected to histological examination. The deparaffinized sections were blocked with 2% BSA for 30 min at 37 °C. Then, sections were incubated with the primary antibody and with the corresponding secondary antibody overnight at 4 °C. Photomicrographs were digitally captured under a Leica DM IRB fluorescence microscope (Leica Microsystems, Philadelphia, PA, USA).

### 4.4. Western Blot Analysis

Bone marrow cells from mice were lysed, and the total protein was extracted. Aliquots of protein (50 μg) or plasma (2 μL) were separated on 12% SDS gels and then transblotted onto a PVDF membrane (Millipore, Bedford, MA). After being blocked with 5% skim milk, the blotting membrane was incubated with the primary antibodies, followed by the corresponding horseradish peroxidase-conjugated secondary antibodies. Bands were visualized using an enzyme-linked chemiluminescence detection kit (PerkinElmer, Waltham, MA, USA), and the band density was measured using quantitative software (TotalLab, Newcastle upon Tyne, UK).

### 4.5. Tail Bleeding Assay

Mice were anesthetized using pentobarbital (50 mg/kg) and then positioned horizontally on a platform to let the tail hang down naturally. A distal 3-mm segment of the tail was cut off with a scalpel. To evaluate bleeding from the incision, filter paper (Kotobuki, Tokyo, Japan) was used to touch the edge of the forming clot every 30 s over a period of 600 s. The dried filter paper was scanned to measure the blood loss. To dissolve the hemoglobin, the filter papers were dipped into 0.04% NH_4_OH solution for 2 h. The level of hemoglobin was assessed by a microplate spectrophotometer (Molecular Devices, Sunnyvale, CA, USA) at 416 nm. The volume of blood in each sample was calculated as compared to the standard curve, which was obtained by dropping defined volumes of WT mouse blood onto a filter paper and extracting hemoglobin as described above.

### 4.6. Coagulation Test

The activation of extrinsic and intrinsic coagulation pathways was examined using PT, aPTT and FIB assay kits according to the manufacturer’s instructions. The activity of PT, aPTT and FIB was assessed using an analyzer (TECO Medical Instruments, Neufahrn NB, Germany).

### 4.7. Thromboelastography (TEG)

Mice were killed by CO_2_, and blood samples were isolated by cardiac puncture. For TEG analysis, 340 μL citrated blood was mixed with 20 μL CaCl_2_ (0.2 M) and then subjected to the Heamoscope TEG 5000 Thrombelastograph Hemostasis Analyzer (Haemonetics Corp., Braintree, MA, USA) according to the manufacturer’s instructions. The major coagulation parameters of TEG were assessed, including *R*, the time to the formation of initial fibrin threads; *K*, the time until the clot reaches a certain strength; α angle, the rapidity with which the clot forms; maximum amplitude, the clot’s maximum strength; amplitude, the end time amplitude; and coagulation index, the overall clotting activity. 

### 4.8. Statistical Analysis

The results are presented as the mean ± SEM. The Mann–Whitney U test was used to compare two independent groups. Differences were considered statistically significant at *p* < 0.05. The Shapiro–Wilk test was used for testing normality. The *p*-value for each group was at *p* > 0.05, and there was no significant difference. SPSS software v8.0 (SPSS Inc., Chicago, IL, USA) was used for all statistical analyses.

## Figures and Tables

**Figure 1 ijms-23-02064-f001:**
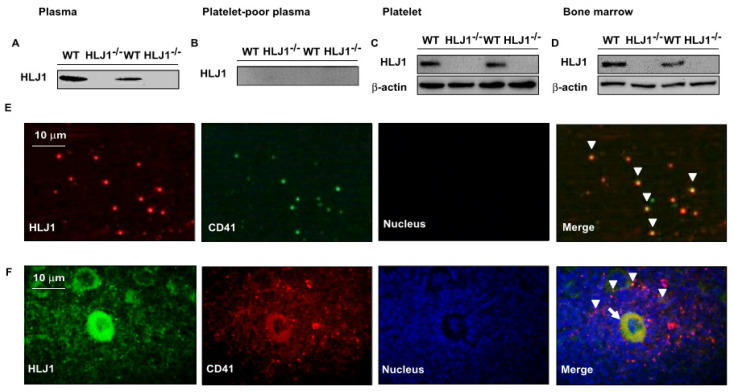
Expression of HLJ in plasma and bone marrow of WT and HLJ1^-/-^ mice. (**A**–**D**) Western blot analysis of HLJ1 in plasma, platelet-poor plasma, platelets and bone marrow. (**E**) Representative images of HLJ1 expression in platelets (arrowheads) of plasma by immunohistochemistry staining. (**F**) Representative images of HLJ1 expression in platelets (arrowheads) and megakaryocytes (arrow) of bone marrow by immunohistochemistry staining.

**Figure 2 ijms-23-02064-f002:**
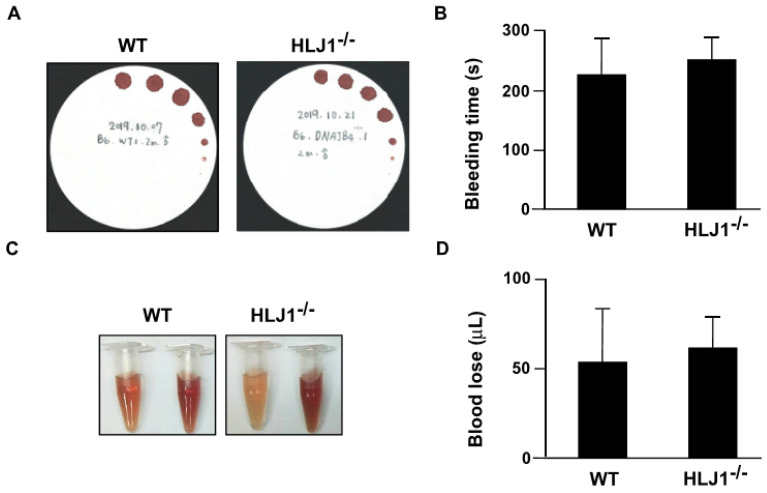
Tail bleeding time and blood loss in WT and HLJ1^-/-^ mice. (**A**,**B**) Tail bleeding time was assessed by the filter paper method in WT and HLJ1^-/-^ mice. (**C**,**D**) Blood loss after tail transection was measured in WT and HLJ1^-/-^ mice. Data are the mean ± SEM from 9 mice.

**Figure 3 ijms-23-02064-f003:**
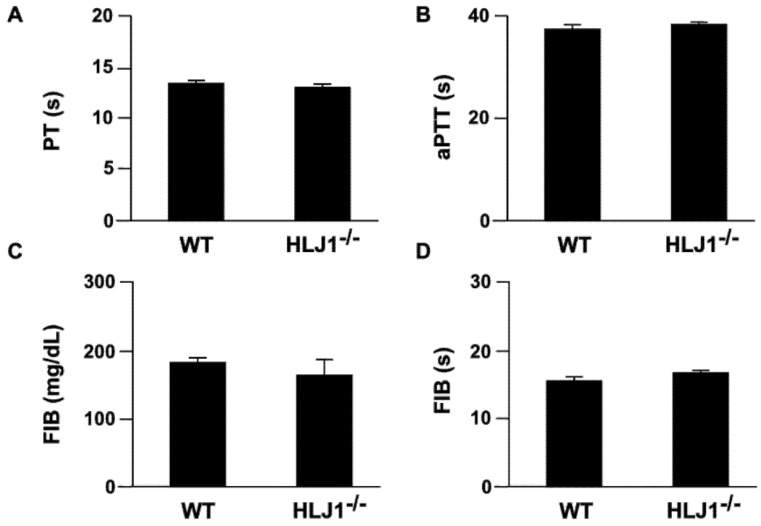
The parameters of plasma coagulation in WT and HLJ1^-/-^ mice. (**A**,**B**) The plasma of mice was isolated, and prothrombin time (PT) and activated partial thromboplastin time (aPTT) were evaluated. (**C**,**D**) The plasma level of fibrinogen (FIB) and time of fibrin clot formation (s) were assessed. Data are the mean ± SEM from 9 mice.

**Figure 4 ijms-23-02064-f004:**
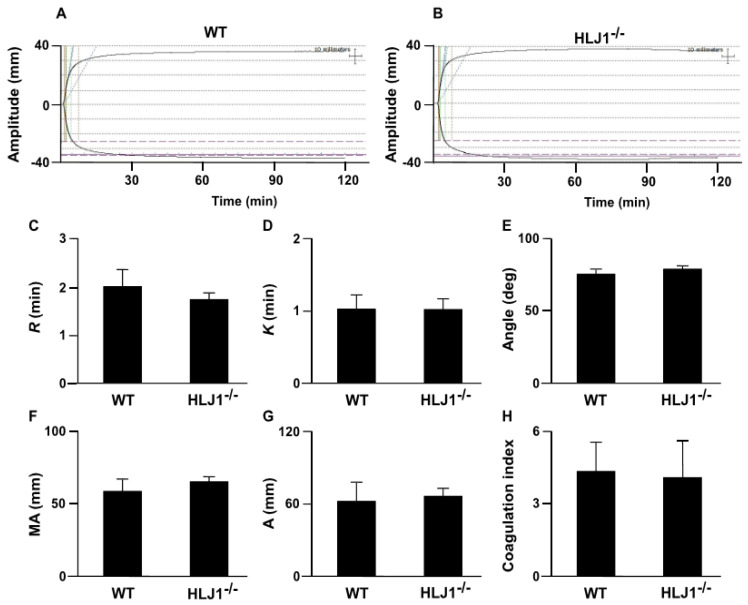
Thromboelastography (TEG) tracing in WT and HLJ1^-/-^ mice. (**A**,**B**) Representative tracing depicting the rate of clot formation and degradation was performed in WT and HLJ1^-/-^ mice. (**C**–**H**) The reaction time (*R*), kinetics time (*K*), α angle, maximum amplitude (MA), the end time amplitude and the coagulation index (**A**) were analyzed. Data are the mean ± SEM from 9 mice.

**Table 1 ijms-23-02064-t001:** Body and tissue weights of WT and HLJ1^-/-^ mice.

Organ Weight	WT (*n* = 9)	HLJ1^-/-^ (*n* = 9)	*p*-Value
Body weight	22.61 ± 0.765	24.68 ± 0.635 *	0.0236 *
Heart	0.17 ± 0.003	0.20 ± 0.002 *	0.0033 *
Liver	1.24 ± 0.072	1.29 ± 0.095	0.453
Spleen	0.06 ± 0.001	0.06 ± 0.009	0.822
Lung	0.13 ± 0.005	0.12 ± 0.004	0.247
Kidney	0.30 ± 0.014	0.35 ± 0.038	0.375
Brown adipose tissue	0.04 ± 0.004	0.05 ± 0.006	0.315
White adipose tissue	0.48 ± 0.054	0.64 ± 0.036 *	0.004 *
Gastrocnemius muscle	0.25 ± 0.015	0.24 ± 0.011	0.871
Brain	0.45 ± 0.03	0.45 ± 0.004	0.470
**The ratio of organ weight to body weight**			
Heart (g/BW)	0.0075 ± 0.000144	0.0081 ± 0.000112 *	0.0253 *
Liver (g/BW)	0.0548 ± 0.00255	0.0522 ± 0.00409	0.199
Spleen (g/BW)	0.0024 ± 0.0000422	0.0024 ± 0.0000881	0.273
Lung (g/BW)	0.0056 ± 0.00021	0.0055 ± 0.000298	0.803
Kidney (g/BW)	0.0132 ± 0.000480	0.0141 ± 0.00121	0.453
Brown adipose tissue (g/BW)	0.0017 ± 0.000164	0.0019 ± 0.0004	0.977
White adipose tissue (g/BW)	0.0212 ± 0.00008	0.0259 ± 0.00028 *	0.0359 *
Gastrocnemius muscle (g/BW)	0.0110 ± 0.000508	0.0097 ± 0.000683	0.309
Brain (g/BW)	0.0199 ± 0.000232	0.0189 ± 0.000482	0.15

Data are the mean ± SEM from 9 mice. * *p* < 0.05 compared to WT mice.

**Table 2 ijms-23-02064-t002:** Peripheral blood counts of WT and HLJ1^-/-^ mice.

Peripheral Blood Counts	WT (*n* = 5)	HLJ1^-/-^ (*n* = 5)	*p*-Value
RBC (×10^6^/mL)	10.82 ± 0.21	10.49 ± 0.088	0.236
PLT (×10^3^/mL)	922.00 ± 42.02	1074.33 ± 83.73	0.176
WBC (×10^3^/mL)	6.58 ± 0.71	5.29 ± 0.60	0.296
NEUT (×10^3^/mL)	0.47 ± 0.11	0.41 ± 0.10	0.703
LYMPH (×10^3^/mL)	6.00 ± 0.58	4.81 ± 0.49	0.474
MONO (×10^3^/mL)	0.04 ± 0.002	0.03 ± 0.008	0.338

RBC: red blood cell; PLT: platelet; WBC: white blood cell; NEUT: neutrophil; LYMPH: lymphocyte; MONO: monocyte. Data are the mean ± SEM from 5 mice.

## Data Availability

The data that support the findings of this study are available from the corresponding author upon reasonable request.

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
