# Peer review of "Genetic Deletion of HLJ1 Does Not Affect Blood Coagulation in Mice"

_ijms, 2022, doi:10.3390/ijms23042064_

Round 1

Reviewer 1 Report

I am happy to provide my opinion on the work of Man-Chen Hsu et al. 

Here are few suggestions:

The results section should not consist parts of the methodology section (example 2.1 introduction etc.) but only results should be reported. Please revise.
Table 1 and 2 - please report p value for all lines.

No suggestions should be reported in the results section, but in discussion (example *These findings suggest that...*. Please revise 
Figure 1: *Data are the mean ± SEM from 9 mice. *P<0.05 vs. WT mice.* on which P value are you referring? Where are mean and EM presented?

4.8. Statistical analysis: Please provide results of data-normality analysis for all cases where Mann-Whitney U test was used

Acceptance could be considered after implementation of mentioned suggestions

Author Response

Reviewer #1

  1. The results section should not consist parts of the methodology section (example 2.1 introduction etc.) but only results should be reported. Please revise.

Response: We fully agree with the reviewer’s viewpoint. In response to the reviewer’s suggestion, we have deleted the descriptions about methodology throughout the section of results section in our revised manuscript.

  1. Table 1 and 2 - please report p value for all lines.

Response: We fully agree with the reviewer’s viewpoint. In response to the reviewer’s suggestion, we have reported p value for all lines in our revised Table 1 and 2.

  1. No suggestions should be reported in the results section, but in discussion (example *These findings suggest that...*. Please revise. Figure 1: *Data are the mean ± SEM from 9 mice. *P<0.05 vs. WT mice.* on which P value are you referring? Where are mean and EM presented?

Response: We thank the reviewer for the professional suggestion. In response to the reviewer’s suggestion, we have revised our result section. In addition, the description about the statistical analysis in Figure 1 has been deleted in our revised manuscript.

  1. 4.8. Statistical analysis: Please provide results of data-normality analysis for all cases where Mann-Whitney U test was used

Response: We thank the reviewer for reminding us this important issue. In response to the reviewer’s suggestion, we have described the statistical analysis for data-normality analysis in the section of Statistical analysis.

Reviewer 2 Report

Hsu et al address the role of the Hsp40 chaperone HLJ1 in blood coagulation using a mouse model deficient in this protein. There is increasing support for important roles of HSPs in haemostasis and the study by Hsu et al. is clearly justified. Moreover, if I’m correct, this seems to be the first study that uses a knockout mouse model to address the role of a HSP haemostasis, as studies so far have made use of inhibitors.

The results presented in this study, although negative, are an important contribution and deserve publication. The minus side and my main critique is that, by restricting themselves to one aspect of haemostasis (namely coagulation), the authors are missing an opportunity to explore roles in platelet function. This would be a cleaner approach than using inhibitors (e.g. Jackson et al, JTH 2020), that are never totally specific, and would have made for a more comprehensive study. Arguably, the tail bleeding test and the thromboelastography assay combine both aspects, but, still, the contribution of HLJ1 to platelet function needs to be investigated in isolation.

Major:

-Information about the mouse model should be provided. If the mouse has already been published elsewhere, please provide a reference. Otherwise details on the generation and genotyping of the strain should be included.

-Table 1. Organ weights should be expressed normalized to the corresponding body weight and the statistical analysis repeated. It is very likely that the apparent cardiac hypertrophy is not such.

-The claim that HLJ1 is expressed in platelets needs to be supported by a western blot of platelet lysates as well as an immunostaining of platelets isolated from blood.

-Figure 1, panel C is very difficult to interpret. Can the authors add some arrows etc to point at relevant features? It would be good to see something that clearly looks like a megakaryocyte.

-The presence of HLJ1 at high levels in plasma is intriguing, as this would argue towards roles of this protein both intracellularly (in the platelet) and in circulation. The mechanisms are expected to be fundamentally different; however, this circulating chaperone aspect is not commented on at all in the manuscript.

-Fig. 4, panels A and B, the graphs should have fully labelled axes.

Minor:

-Revise the usage of the English language throughout and in particular the abstract, e.g. “we investigated the role”, not “We investigated that the role”.

-Check the last two sentences of the first from last paragraph of the introduction (“While the correlation…”). The authors probably mean: While the role of HLJ1 in cancer biology has been extensively investigated, their significance in blood coagulation is still elusive.

-Remove the sentence “Two-month-old male C57BL/6 WT…..” from the figure legends, this information is already in the methods section.

-Table 2, express counts as (x106/µl) or (x103/µl) rather than using M and K. Ensure you use µ rather than u. In the legend remove *P<0.05 etc, as there are no significant differences.

-Fig. 2, are panels A and C representative of one experiment? Please indicate so.

-Fig. 4, legend, change to: (A and B) Representative tracing…

-Discussion, first paragraph, replace “illuminating” by a more appropriate term like “conclude”

Author Response

Reviewer #2

Major:

  1. Information about the mouse model should be provided. If the mouse has already been published elsewhere, please provide a reference. Otherwise details on the generation and genotyping of the strain should be included.

Response: We thank the reviewer for reminding us this important issue. In response to the reviewer’s suggestion, we have added the description about the generation of HLJ1 KO mice in the section of methodology in our manuscript. Now it read as “HLJ1-/- C57BL/6 mice were generated by the gene targeting strategy. Specifically, exon2 of mouse HLJ1 was deleted by traditional homologous recombination in embryonic stem cells. HLJ1-/- mice were generated and the polymerase chain reaction (PCR) of genomic DNA was performed to confirm HLJ1 −/− genotype. PCR was performed with the primers: HLJ1 (forward): 5’- GGC TTG CTG TCT AAG GTG ATG -3’, HLJ1 (reverse): 5’- ACG TTC AAA GCT ATC ATA GTC -3’ at 95 °C for 5 min, followed by 40 cycles at 95 °C for 30 s, 62 °C for 30 s, 72 °C 30 s and additional extension at 72 °C for 5 min”. (page 14, line 253-260).

  1. Table 1. Organ weights should be expressed normalized to the corresponding body weight and the statistical analysis repeated. It is very likely that the apparent cardiac hypertrophy is not such.

Response: We fully agree with the reviewer’s viewpoint. In response to the reviewer’s suggestion, we have reported the results of the ratio of organ weight to body weight in our revised manuscript.

  1. The claim that HLJ1 is expressed in platelets needs to be supported by a western blot of platelet lysates as well as an immunostaining of platelets isolated from blood.

Response: We thank the reviewer for reminding us this important issue. We have performed additional experiments to examine the localization of HLJ1 in plasma. Our new data showed that no soluble form of HLJ1 was detected in platelet-poor plasma; in contrast, the protein expression of HLJ1 in plasma was mainly restricted in platelets. In response to the reviewer’s suggestion, we have reported these new data in figure 1 of our revised manuscript. Accordingly, the section of results has also been revised (page 6, line 109-113).

  1. Figure 1, panel C is very difficult to interpret. Can the authors add some arrows etc. to point at relevant features? It would be good to see something that clearly looks like a megakaryocyte.

Response: We fully agree with the reviewer’s viewpoint. In response to the reviewer’s suggestion, we have added arrowheads and arrow to label platelet and megakaryocytes, respectively. Accordingly, figure legend of figure 1 has also been revised.

  1. The presence of HLJ1 at high levels in plasma is intriguing, as this would argue towards roles of this protein both intracellularly (in the platelet) and in circulation. The mechanisms are expected to be fundamentally different; however, this circulating chaperone aspect is not commented on at all in the manuscript.

Response: The reviewer is right. The roles of HLJ1 both intracellularly (in the platelet) and in circulation should be different. However, our new data found that HLJ1 is mainly appeared in the platelets but not as a soluble protein in circulation. these new data in figure 1 of our revised manuscript.

  1. Fig. 4, panels A and B, the graphs should have fully labelled axes.

Response: We thank the reviewer for reminding us this important issue. The x axes mean time (min) and y axes mean amplitude (mm) in figure 4. In response to the reviewer’s suggestion, the axes in panels A and B of figure 4 have be defined in our revised manuscript.

Minor:

  1. Revise the usage of the English language throughout and in particular the abstract, e.g. “we investigated the role”, not “We investigated that the role”.

Response: In response to the reviewer’s suggestion, we have corrected this error.

  1. Check the last two sentences of the first from last paragraph of the introduction (“While the correlation…”). The authors probably mean: While the role of HLJ1 in cancer biology has been extensively investigated, their significance in blood coagulation is still elusive.

Response: We fully agree with the reviewer’s viewpoint. In response to the reviewer’s suggestion, we have revised this sentence in the section of introduction.

  1. Remove the sentence “Two-month-old male C57BL/6 WT….” from the figure legends, this information is already in the methods section.

Response: We fully agree with the reviewer’s suggestion, we removed the sentence “Two-month-old male C57BL/6 WT….” from the figure legends in figure 1-4 of our revised manuscript.

  1. Table 2, express counts as (x106/µl) or (x103/µl) rather than using M and K. Ensure you use µ rather than u. In the legend remove *P<0.05 etc, as there are no significant differences.

Response: We fully agree with the reviewer’s viewpoint. In response to the reviewer’s suggestion, we have revised the Table 2, we replaced M and K with (x106/µl) or (x103/µl) and removed the description about “*P<0.05 etc., as there are no significant differences”.

  1. Fig. 2, are panels A and C representative of one experiment? Please indicate so.

Response: The reviewer’s is right; the panels A to D represent one experiment for evaluating the blood loss. Panel C is the representative image for the dissolution of blood dots in panel A, which is used for quantifying the blood loss in Panel D.

  1. Fig. 4, legend, change to: (A and B) Representative tracing…

Response: We have changed (A and B) TEG tracing to (A and B) Representative tracing… in figure 4 legend and marked in red in our revised manuscript.

  1. Discussion, first paragraph, replace “illuminating” by a more appropriate term like “conclude”

Response: In response to the reviewer’s suggestion, we have replaced the word “illuminating” with “concluded” and marked in red in our revised manuscript (page11, line 175).

Round 2

Reviewer 1 Report

Dear Authors,

Thank you for implementing most of the recommendations. This work is still missing a good scientific writting.

Author Response

We thank the reviewer for the positive feedback for our revised manuscript.  In response to the issue about scientific writing suggested by the reviewer, we have submitted our manuscript to the service of MDPI English editing.  We hope that the our revision can meet your expection.

Reviewer 2 Report

The authors have address my concerns satisfactorily.

Author Response

We thank the reviewer for the positive feedback for our revised manuscript.